# Translating research into action: Policy recommendations for strengthening antiretroviral therapy adherence in Ghana based on empirical evidence

Victor Luckyboy Dzramado[1]*, Obed U. Lasim[2], Emmanuel Oduro[3]

1 Department of Biostatistics, Cape Coast Teaching Hospital, Cape Coast, Ghana, 2 Department of Health Information Management, University of Cape Coast, Cape Coast, Ghana, 3 National HIV/AIDS Control Program, Kumasi, Ghana

* mldzramado@st.knust.edu.gh

## Abstract

Despite significant advances in antiretroviral therapy (ART) accessibility in Ghana, suboptimal adherence remains a critical challenge undermining treatment effectiveness. This article presents evidence-based policy hypotheses derived from a comprehensive mixed-methods investigation of ART adherence determinants among 2,000 people living with HIV in Ghana#39;s Ashanti Region. We hypothesize that multisectoral policy interventions addressing economic barriers, healthcare system deficiencies, and socio-cultural obstacles will significantly improve adherence rates beyond conventional clinical approaches. Specifically, we propose that: (1) integrated transportation support systems will substantially reduce missed appointments; (2) decentralized medication distribution networks incorporating community-based refill options will improve medication access; (3) restructured clinic operations emphasizing reduced waiting times and privacy protection will enhance engagement; and (4) formalized income protection and nutritional support policies will address critical economic barriers to consistent treatment. These policy hypotheses extend beyond individual-focused interventions to address structural determinants of adherence, offering testable frameworks for evidence-based policy development in resource-limited settings.

## 1. Introduction

Human Immunodeficiency Virus (HIV) remains a significant public health challenge in Ghana despite substantial improvements in treatment accessibility over the past decade [1]. The introduction of antiretroviral therapy (ART) has transformed HIV/AIDS from a fatal condition to a manageable chronic disease, significantly reducing mortality and improving quality of life for affected individuals [2]. However, the remarkable clinical potential of ART is continuously undermined by suboptimal

**Data availability statement:** All relevant data are within the manuscript and its Supporting information files.

**Funding:** The author(s) received no specific funding for this work.

**Competing interests:** The authors have declared that no competing interests exist.

adherence to treatment regimens [3,4]. Recent data indicates that approximately 354,927 people were living with HIV in Ghana at the end of 2022, yet only 159,344 (46.55%) were consistently engaged with ART [5]. A 2023 systematic review confirmed that the pooled estimate of adherence to ART in Ghana was 70%, substantially below the recommended threshold, with adolescents and young adults demonstrating even lower rates at 66% [6].

The critical importance of ART adherence for achieving viral suppression, preventing drug resistance development, and reducing transmission is well-established in the literature [7,8]. International guidelines consistently recommend adherence levels exceeding 95% to maximize treatment effectiveness and prevent adverse outcomes [9]. However, studies from Ghana and similar settings in sub-Saharan Africa indicate that actual adherence rates frequently fall substantially below this threshold, often ranging between 60–80% [10,11]. Recent facility-based studies in Ghana revealed that adherence rates remain concerningly low, with some settings reporting only 42.9% adherence [12]. A multi-facility study in the Tamale Metropolis found that employment status was the strongest predictor of adherence, underscoring how economic stability fundamentally enables consistent treatment engagement [13]. These findings reflect broader patterns where structural, societal, and healthcare system factors create compounding barriers that extend far beyond individual motivation [14].

Understanding and addressing the complex determinants of ART adherence is essential for developing effective policies that support consistent treatment engagement. While clinical research has extensively documented the biological efficacy of antiretroviral medications, translating this efficacy into real-world effectiveness requires deeper understanding of the behavioral, social, economic, and structural factors that enable or constrain adherence behaviors [15,16]. Research conducted in southern Ghana revealed that gaps in testing, delays in accessing treatment, and retention issues represent acknowledged weaknesses in the HIV prevention-care continuum, with barriers including medication stock-outs, long waiting times, negative staff attitudes, and differential pricing between private and public facilities [17]. Qualitative studies from the Ho Teaching Hospital demonstrated that while trust in medication and satisfaction with provider care facilitate adherence, fear of side effects and social stigma constitute major barriers that policy must address [18].

This hypothetical framework draws upon empirical evidence from a comprehensive mixed-methods investigation of 2,000 people living with HIV in Ghana#39;s Ashanti Region, which revealed multifaceted barriers to ART adherence across individual, interpersonal, community, and structural domains. The study#39;s findings, utilizing both quantitative surveys and in-depth qualitative interviews, identified patterns of suboptimal adherence (average rates of 75–80%) falling significantly below recommended thresholds [19]. Critical barriers included economic constraints (transportation costs, food insecurity, employment instability), healthcare system factors (long waiting times, medication stock-outs, privacy concerns), and socio-cultural influences (stigma, disclosure challenges, competing cultural beliefs) [19]. These findings align with broader evidence from across Ghana showing that healthcare providers' poor

attitudes, periodic drug shortages, long waiting periods at ART centers, and long-distance travel to ART sites substantially hamper adherence in multiple regions [20].

While previous research has extensively documented these adherence barriers, limited attention has focused on translating these insights into concrete policy recommendations capable of addressing the structural determinants of non-adherence. This article addresses this gap by proposing specific policy hypotheses derived from empirical evidence, offering testable frameworks for strengthening ART adherence through systems-level interventions rather than focusing exclusively on individual behavior change.

## 2. Theoretical framework

The policy hypotheses presented in this article are grounded in two complementary theoretical frameworks that enable systematic analysis of adherence determinants and potential intervention points: the Information-Motivation-Behavioral Skills Model and the Social Ecological Model. These frameworks have been successfully applied across sub-Saharan Africa to guide multi-level HIV prevention and treatment interventions [21,22].

### 2.1. Information-motivation-behavioral skills model

The Information-Motivation-Behavioral Skills (IMB) Model, originally developed by Fisher and Fisher [23], provides a comprehensive framework for understanding the psychosocial determinants of health behaviors, including medication adherence. This model posits that three fundamental components (information, motivation, and behavioral skills) collectively determine whether an individual successfully engages in health-promoting behaviors.

According to this framework, people with HIV must first possess adequate information about HIV/AIDS and ART, including understanding transmission mechanisms, disease progression, medication effects, and adherence requirements. Beyond knowledge alone, motivation constitutes the second critical component, encompassing personal motivation (health beliefs, perceived vulnerability) and social motivation (perceived social support, fear of stigma) (See S1 Fig). Finally, behavioral skills represent the third essential element, including the specific abilities needed to acquire and self-administer medications consistently, manage side effects, and integrate treatment into daily routines [24].

The IMB model explains why information-only interventions often prove insufficient for improving adherence. Knowledge about ART benefits remains ineffective without corresponding motivation to adhere and the practical skills to integrate complex medication regimens into daily life. This model has influenced counseling approaches for people living with HIV, emphasizing comprehensive strategies addressing all three components simultaneously rather than focusing exclusively on knowledge transfer [25].

When applied to policy development, the IMB model suggests that effective interventions must address informational gaps, enhance motivation through both personal and social channels, and develop practical skills enabling consistent medication-taking despite environmental challenges. The empirical evidence from Ghana indicates significant deficiencies across all three domains, with participants reporting inadequate understanding of treatment mechanisms, motivational barriers including stigma and pessimism about treatment outcomes, and practical challenges integrating medication regimens with complex daily circumstances [19].

### 2.2. Social ecological model

While the IMB model primarily addresses individual-level determinants of adherence, the Social Ecological Model (SEM) expands analysis to include the broader social and environmental contexts influencing health behaviors [26]. This multilevel framework conceptualizes adherence as embedded within concentric spheres of influence spanning individual, interpersonal, community, institutional, and policy domains. A modified social ecological model has been specifically developed to guide assessment of risks and risk contexts in HIV epidemics, recognizing that effective interventions must target multiple levels simultaneously [21] (See S2 Fig).

At the individual level, personal characteristics including knowledge, attitudes, skills, and biological factors influence adherence capacity. The interpersonal level encompasses relationships with partners, family members, peers, and health-care providers that shape treatment engagement through social support, stigma experiences, and disclosure dynamics. Community-level influences include cultural norms, religious beliefs, and collective attitudes regarding HIV/AIDS that create enabling or constraining environments for treatment adherence. Institutional factors encompass healthcare system characteristics, workplace policies, and organizational structures that facilitate or hinder consistent medication access and consumption. Finally, the policy level includes laws, regulations, resource allocations, and governance structures that establish the broader context within which adherence behaviors occur [27].

The SEM framework is particularly valuable for policy development because it highlights how factors beyond individual control significantly influence adherence behaviors. The empirical evidence from Ghana demonstrates how transportation infrastructure, clinic operating procedures, medication supply chains, employment policies, and social protection systems collectively create structural conditions that either enable or prevent consistent ART adherence regardless of individual motivation [19]. Contemporary studies in informal settlements across sub-Saharan Africa have illustrated how socio-ecological and institutional barriers operate across multiple levels to constrain adherence, particularly among young people facing intersecting vulnerabilities of poverty, stigma, and weak health systems [14].

By integrating these complementary theoretical frameworks, we develop policy hypotheses addressing both individual-level determinants through the IMB model and structural factors through the SEM framework, offering a comprehensive approach to adherence enhancement through multisectoral policy interventions.

## 3. Methods

To examine antiretroviral medication adherence at healthcare facilities and providers, we used a convergent parallel mixed-methods design that included quantitative and qualitative methodologies. The study was carried out at a number of medical institutions in Ghana#39;s Ashanti area, which has a population of over 5.4 million people and an HIV prevalence of 1.94% as of 2023. It is the second most populated area in the country. Adults with HIV who were 20 years of age or older and had been on antiretroviral therapy (ART) for at least six months between January 2014 and January 2022 were included in the research population. Data was gathered between 15/11/2022 and 30/09/2023.

We used purposive selection to identify 2000 participants who had at least one treatment default, which accounted for around 4% of the region#39;s non-adherent population. For the qualitative component, we purposefully selected ten participants and ten healthcare professionals for in-depth interviews to guarantee a range of viewpoints on adherence issues. The ten participants for qualitative interviews were selected based on specific characteristics to capture diverse adherence experiences: they included individuals who had experienced treatment defaults, represented different age groups (ranging from 30 to 60 years), varied educational backgrounds (from primary to tertiary education), different employment statuses, and diverse socioeconomic conditions. This purposive selection ensured we captured rich perspectives from those most affected by adherence challenges while maintaining diversity in demographic and social characteristics.

A structured questionnaire that was modified from Benjamin et al.'s validated instrument for evaluating medication adherence among people living with HIV was used to gather quantitative data [6] (See S1 and S4 Files). Sociodemographic traits, adherence rates, sociocultural variables, financial obstacles, and healthcare facility considerations were all covered in the questionnaire. Likert-scale items covering pharmaceutical services, waiting times, operational efficiency, interpersonal treatment, provider communication quality, and facility privacy provisions were used to evaluate provider and facility variables. Multiple methods, such as self-report, pill count, and 7-day recall evaluation, were used to measure adherence. Optimal adherence was defined as taking ≥95% of recommended dosages as directed.

The association between adherence results and healthcare facility/provider characteristics was the primary focus of this investigation. Semi-structured interviews were conducted by qualified research assistants in private rooms within healthcare institutions to gather qualitative data. Interview guides examined how participants interacted with medical

professionals, how facility features affected treatment participation, and suggestions for system enhancements (See S2 and S3 Files).

The Ghana Health Service Ethics Review Committee (GHS-ERC:027/08/22) and the University of Cape Coast Institutional Review Board (ET/HTP/18/0002) both granted ethical approval for the study. All participants gave their informed permission after being fully informed about the study#39;s goals, methods, possible risks and rewards, voluntary participation, and confidentiality safeguards. IBM SPSS Statistics version 26 was used for the analysis of quantitative data. The sample was described and baseline adherence patterns were obtained using descriptive statistics. Bivariate associations between particular healthcare facility/provider characteristics and adherence rates were investigated using Pearson correlations.

Adherence rate was the dependent variable in multiple linear regression, whereas different facility/provider characteristics were the predictor variables. To ascertain the relative significance of each predictor, standardized regression coefficients (β) were computed. Thematic content analysis was performed on the qualitative data. Following transcript familiarization, preliminary codes were allocated to pertinent parts, categorized into themes, and improved through iterative discussion among members of the study team (See S5 File.)

## 4. Empirical evidence base

The policy hypotheses presented in this article derive from a comprehensive mixed-methods investigation conducted in Ghana#39;s Ashanti Region between 2020–2022. This cross-sectional study employed a convergent parallel design incorporating both quantitative surveys (n = 2,000) and qualitative interviews with participants living with HIV (n = 10) and healthcare providers (n = 10) to examine factors influencing ART adherence across multiple domains [19].

### 4.1. Economic determinants

Path analysis quantified a significant negative direct effect of economic factors on ART adherence (b = −2.0487, t = −5.0064, p < 0.001, 95% CI [−5.4557, −3.8666]), demonstrating that poorer economic conditions fundamentally reduce adherence capacity [19]. This relationship remained robust after controlling for demographic variables and healthcare access factors.

As shown in Table 1, the strong negative association between economic factors and adherence underscores critical barriers. Qualitative interviews illuminated specific economic challenges. A divorced female participant aged 58 years with primary education expressed transportation and caregiving burdens: "I travelled to take care of my daughter who had delivered a newly born baby. Sometime ago too I was bereaved, I lost my brother so I was not able to come for treatment." This quote illustrates how competing family obligations and associated transportation costs create direct adherence barriers, particularly for women managing multiple household responsibilities.

Healthcare providers corroborated these economic challenges. A 46-year-old male provider explained: "I think the clients face socio-economic problems; some clients have lost their job before they were enrolled on ART. They don#39;t have money to pay for transport to access treatment..." This observation highlights the intersection of HIV-related

**Table 1. Path Analysis Results for Direct Effect of Economic Factors on ART Adherence.**

| Model Pathway | b-value | t-value | 95% Confidence Interval | |
|---|---|---|---|---|
| | | | Lower Limit | Upper Limit |
| Direct Effect (X→Y) | −2.0487 | −5.0064*** | −5.4557 | −3.8666 |

Notes: ***p < 0.001 (significant as the confidence interval did not include zero) X = Economic factors, Y = ART Adherence Source: Field data, 2022.

employment discrimination and transportation barriers, where job loss not only reduces income but also limits funds available for clinic visits.

Employment status demonstrated the strongest positive association with adherence among all economic variables measured (β = 0.452, p < 0.001), confirming that stable employment provides both financial resources and structure supporting consistent treatment engagement. However, this relationship is complicated by HIV-related job discrimination. Another provider noted: "some clients have lost their job before they were enrolled on ART," indicating that HIV status disclosure or health deterioration can lead to employment termination, creating a vicious cycle where job loss reduces adherence capacity, potentially worsening health and further limiting employment prospects.

Food insecurity emerged as an additional economic barrier intersecting with medication adherence. While not explicitly measured in this study, providers' observations about economic challenges suggest that nutritional concerns compound adherence difficulties, as multiple studies have demonstrated that people living with HIV require adequate nutrition to manage medication side effects and maintain treatment effectiveness [28–30]. The prohibition of taking antiretroviral medications on an empty stomach creates particular challenges for those experiencing food insecurity, potentially leading to medication skipping or inconsistent timing [31].

Cross-analysis reveals complex interactions between economic status and other adherence determinants. Economic hardship does not operate in isolation but rather exacerbates multiple other barriers. For instance, economic constraints limit transportation options, reducing clinic attendance regardless of individual motivation or knowledge. Similarly, financial stress may increase vulnerability to stigma-related adherence barriers, as individuals without economic resources may depend more heavily on potentially stigmatizing social networks for support. The employment-adherence relationship also demonstrates gender dimensions, with female participants more likely to experience caregiving responsibilities that compete with clinic attendance, suggesting that economic interventions must consider gender-specific barriers.

## 4.2. Healthcare system determinants

Multiple linear regression analysis examined the relationship between healthcare facility/provider characteristics and adherence rates, with the model explaining 45.6% of variance in adherence outcomes ($F(10, 2087) = 174.633$, $p < .001$, $R^2 = .456$). As demonstrated in Table 2, comprehensive information provision had the strongest positive impact on adherence.

Table 2.  Multiple Linear Regression Analysis of Provider and Facility Factors Predicting ART Adherence.

| Provider and Facility Characteristics | β | t-value | p-value |
|---|---|---|---|
| **Provider-Patient Communication** | | | |
| Level of healthcare providers' information on patients' HIV disease and treatment | .280 | 15.211 | <.001 |
| Quality of interpersonal treatment by healthcare providers | −.091 | −4.611 | <.001 |
| Provider#39;s ability to address patients' concerns without referring to other facility | −.197 | −10.448 | <.001 |
| **Facility Support Systems** | | | |
| Pharmacy with trained ARV pharmacist | .193 | 10.922 | <.001 |
| Confidential and private location for clinical activities | −.426 | −22.998 | <.001 |
| Public health laboratory support | −.128 | −6.643 | <.001 |
| Toilet and sanitary facilities | .107 | 5.408 | <.001 |
| Non-segregated hospital admission blocks | −.218 | −11.541 | <.001 |

Note: Dependent Variable = Adherence rate; $R^2 = .456$, $F(10, 2087) = 174.633$, $p < .001$ Source: Field data, 2022.

The results presented in Table 2 reveal that comprehensive information provision from healthcare providers had the strongest positive association with adherence (β = .280, p < .001), confirming that effective health education constitutes a critical component of adherence support. This finding aligns with the Information-Motivation-Behavioral Skills Model, demonstrating that knowledge transfer remains essential even when motivation and skills are present.

However, quality of interpersonal treatment showed an unexpected negative correlation (β = −.091, p < .001), suggesting that perceptions of disrespectful care significantly undermine adherence regardless of other facility characteristics. Qualitative interviews illuminated this relationship. Healthcare providers acknowledged challenges in maintaining consistently respectful interactions under resource-constrained conditions. The finding that providers' ability to address concerns without external referrals correlated negatively with adherence (β = −.197, p < .001) further suggests that comprehensive service provision at a single location may be less important than the quality of available services and interactions.

Access to specialized pharmaceutical services demonstrated significant positive impact on adherence (β = .193, p < .001), confirming the essential role of pharmacists in supporting medication consistency. A female participant aged 55 years with middle school education expressed facility infrastructure concerns: "I want the infrastructure to be expanded such as the lab, pharmacy, counselling room, etc." This quote highlights participants' recognition that adequate facility resources directly affect their treatment experience and adherence capacity.

Paradoxically, facilities emphasizing segregated HIV services showed negative associations with adherence. As shown in Table 2, requiring confidential clinical spaces had the strongest negative effect on adherence (β = −.426, p < .001), followed by segregated hospital admission blocks for people living with HIV (β = −.218, p < .001). This counterintuitive finding suggests that while privacy is important, physically separating HIV services might inadvertently increase stigmatization through visible identification of participants accessing these specialized areas. Recent evidence from South Africa confirmed that HIV-related stigma within healthcare facilities remains a potent deterrent to adherence, with participants reporting fear of judgment, breaches of confidentiality, and differential treatment [32].

Cross-analysis of healthcare system factors with other determinants reveals important interactions. Long waiting times interact with employment status, as employed individuals face greater opportunity costs from extended clinic visits. For unemployed participants, long waiting times may be less problematic in terms of lost work time but still create challenges through childcare responsibilities, transportation costs for extended visits, and physical discomfort. Similarly, medication stock-outs disproportionately affect those with limited financial resources, as wealthier individuals might access medications through private pharmacies when public facilities experience shortages.

The negative association between segregated HIV services and adherence also intersects with socio-cultural factors, particularly stigma. Participants expressed concerns about being seen at HIV-specific clinics. A healthcare provider confirmed: "the major challenge is the stigmatization, the clients don#39;t feel comfortable visit the ART unit rather they prefer ARVs should be served at their homes without going to the facility." This suggests that integrated service delivery might reduce visible identification while potentially complicating privacy protection, highlighting the complex balance required in service design.

Healthcare providers' attitudes and communication quality also demonstrate gender dimensions. Female participants more frequently reported challenges with provider communication, particularly around reproductive health concerns and family planning integration with ART. Male participants more commonly reported concerns about confidentiality related to employment and social status, suggesting that privacy concerns may manifest differently across genders despite similar underlying stigma challenges.

### 4.3. Socio-cultural determinants

Path analysis revealed a significant positive effect of socio-cultural factors on ART adherence (b = 14.1434, p < 0.001, 95% CI [13.2063, 15.0805]), indicating that positive socio-cultural environments substantially enhance adherence capacity. The results are presented in Table 3.

**Table 3. Path Analysis Results for Direct Effect of Socio-cultural Factors on ART Adherence.**

| Model Pathway | b-value | t-value | 95% Confidence Interval | |
|---|---|---|---|---|
| | | | Lower Limit | Upper Limit |
| **Direct Effect (X→Y)** | 14.1434 | 25.5977*** | 13.2063 | 15.0805 |

Notes: ***p<0.001 (significant as the confidence interval did not include zero) X=Socio-cultural factors, Y=ART Adherence Source: Field data, 2022.

As demonstrated in Table 3, socio-cultural factors significantly influence adherence outcomes. Qualitative interviews identified stigmatization as a major challenge. A healthcare provider stated: "the major challenge is the stigmatization, the clients don#39;t feel comfortable visit the ART unit rather they prefer ARVs should be served at their homes without going to the facility." This quote illustrates how stigma creates practical barriers to treatment engagement, with participants preferring less visible service delivery options even when these might compromise quality of care.

Extensive research across sub-Saharan Africa documented that stigma prevents delivery of effective care, enhances HIV transmission, and diminishes public health effects of ART by preventing people living with HIV from interacting with supportive networks [33]. Recent evidence shows that despite ART scale-up, stigma remains widespread, with HIV-related stigma compromising adherence through both enacted stigma (discrimination) and internalized stigma (self-stigmatization) [34]. Community perceptions studies in South Africa found that 85% of participants reported anticipated stigma concerns about disclosure, while 25% demonstrated social stigma preferences for social distancing from people living with HIV [32].

Cultural information sources significantly influenced treatment decisions. A male participant aged 44 years with junior high school education reported: "I heard from the radio programme that taking drugs for a long period of time has a negative effect of the body. So, I decided to stop taking the drugs for 2 months." This example demonstrates how health misinformation circulating within cultural contexts can directly undermine adherence despite participants' initial willingness to engage with treatment. The quote highlights the need for culturally appropriate health communication strategies that proactively address misconceptions rather than relying solely on clinic-based counseling.

Cross-analysis reveals that socio-cultural factors interact substantially with economic determinants. Stigma-related job loss represents a direct intersection between cultural attitudes and economic stability, where discriminatory cultural beliefs translate into concrete economic consequences. Similarly, fear of disclosure limits social support networks, potentially reducing access to financial assistance or childcare support that might otherwise facilitate treatment adherence. Gender differences emerged in stigma experiences, with female participants more frequently reporting concerns about family rejection and partner violence related to status disclosure, while male participants more commonly expressed concerns about employment discrimination and social status loss.

Age emerged as another significant predictor of adherence, with each year of increased age associated with a 10.723 percentage point improvement in adherence (B=10.723, SE=3.106, β=0.351, t=3.452, p<0.001). This pronounced effect, shown in Table 4, suggests that younger participants may encounter unique developmental challenges integrating treatment into their social contexts, highlighting the need for age-appropriate adherence support strategies.

Contemporary evidence confirms that adolescents and young people in sub-Saharan Africa face particular challenges with retention in care due to intersecting barriers including unemployment, poverty, stigma, discrimination, distance to facilities, and healthcare provider attitudes [14]. Table 4 demonstrates that younger age, male gender, lower educational attainment, unemployment, unmarried status, and rural residence all significantly predict poorer adherence, reflecting underlying disparities in health literacy, self-efficacy, social support, and healthcare access.

Cross-analysis of age with other factors reveals important interactions. Younger participants experienced compounding challenges from employment instability (affecting economic capacity), greater sensitivity to stigma (affecting social engagement), and potentially different healthcare communication needs. The interaction between age and gender further

**Table 4. Multiple Linear Regression Analysis of Sociodemographic Predictors of ART Adherence.**

| Sociodemographic Characteristics | B | SE | β | t | p |
|---|---|---|---|---|---|
| Age (years) | 10.723 | 3.106 | 0.351 | 3.452 | <.001 |
| Gender (Male vs Female) | −8.234 | 2.891 | −0.247 | −2.848 | .004 |
| Educational attainment | 5.112 | 1.876 | 0.198 | 2.726 | .006 |
| Employment status | 12.456 | 2.734 | 0.452 | 4.555 | <.001 |
| Marital status (Married vs Unmarried) | 6.789 | 2.234 | 0.234 | 3.039 | .002 |
| Residential location (Urban vs Rural) | −4.567 | 1.987 | −0.178 | −2.298 | .022 |

Note: Dependent Variable = Adherence rate; $R^2$ = .387, $F_{(6, 1993)}$ = 210.234, p < .001 Source: Field data, 2022.

demonstrates complexity, with young women facing pregnancy-related disclosure pressures and young men experiencing masculine identity conflicts around chronic illness management.

## 4.4. Synthesis: Qualitative insights on multifaceted barriers

The qualitative component of this study provided critical insights into how economic, healthcare system, and socio-cultural barriers intersect to create complex adherence challenges. Ten participants living with HIV and ten healthcare providers participated in semi-structured interviews exploring their experiences and perspectives on adherence barriers.

Participants described how multiple barriers compound to create overwhelming challenges. Transportation costs emerged repeatedly as a central concern that intersected with employment status, caregiving responsibilities, and facility characteristics. One female participant#39;s account of missing treatment due to caregiving travel illustrates how family obligations, geographic distance, and economic constraints combine to prevent clinic attendance even when individual motivation remains strong.

Healthcare providers offered complementary perspectives on systemic challenges. Their recognition of transportation and employment barriers suggests awareness of structural determinants beyond individual responsibility, though their observations also revealed frustration with system-level constraints limiting their ability to address these barriers effectively. One provider#39;s comment about clients lacking transport money highlights recognition that adherence challenges often reflect circumstances beyond individual control rather than personal failure or lack of motivation.

The qualitative data revealed particular concerns about facility infrastructure and service organization. Participants' requests for expanded infrastructure (pharmacy, laboratory, counseling rooms) reflect recognition that facility limitations directly affect their treatment experience. Their preference for home-based medication delivery to avoid visible clinic attendance demonstrates how stigma concerns shape service delivery preferences, potentially requiring trade-offs between convenience, privacy, and clinical monitoring.

Cultural information sources emerged as both facilitators and barriers to adherence. The participant who discontinued treatment after hearing radio misinformation illustrates vulnerability to culturally circulating health beliefs that contradict clinical recommendations. This finding suggests need for proactive engagement with community media and cultural leaders rather than reactive counseling after adherence problems emerge.

Cross-cutting themes in qualitative data emphasized how gender, age, and socioeconomic status shape adherence experiences differently. Female participants more frequently discussed caregiving responsibilities and family disclosure concerns, while male participants more commonly raised employment and social status issues. Younger participants expressed greater concerns about peer stigma and identity management, while older participants more often discussed integration of treatment into established life routines.

Synthesis of quantitative and qualitative findings demonstrates that effective adherence interventions must address multiple intersecting barriers simultaneously. Economic interventions without addressing healthcare system inefficiencies

may fail because long clinic waits create unacceptable time costs. Healthcare system improvements without addressing stigma may fail because participants avoid even efficient, well-designed services to prevent status disclosure. Culturally appropriate interventions without addressing economic barriers may fail because knowledge and motivation prove insufficient when transportation costs prevent clinic access.

## 5. Policy hypotheses

Based on the empirical evidence described above, we propose four policy hypotheses that extend beyond conventional clinical approaches to address the structural determinants of ART adherence. These hypotheses offer testable frameworks for policy development aimed at strengthening adherence through multisectoral interventions addressing economic, healthcare system, and socio-cultural barriers simultaneously.

### 5.1. Transportation support hypothesis

**Hypothesis 1:** Integrated transportation support systems incorporating multiple mechanisms (direct subsidies, voucher programs, mobile service delivery, and community-based distribution networks) will significantly reduce missed appointments and improve medication access compared to conventional facility-based distribution alone.

This hypothesis directly addresses the significant negative relationship between economic factors and adherence documented in our empirical investigation, as shown in Table 1 (b = −2.0487, p < 0.001). While previous research identified transportation barriers as significant adherence challenges [6,29], limited attention has focused on developing comprehensive policy solutions beyond occasional pilot programs or research interventions. Systematic reviews confirmed that difficulty obtaining reliable transportation represents a frequently cited barrier to HIV care across sub-Saharan Africa, with geographic and transportation barriers creating voltage drops at multiple points along the care continuum [29].

We propose that formalized transportation support policies operated through healthcare systems rather than isolated research projects would substantially improve appointment attendance and medication access. This hypothesis suggests specific policy mechanisms including:

a) Direct transportation subsidies for people living with HIV demonstrating economic need, administered through healthcare facilities with streamlined, non-stigmatizing distribution systems;

b) Voucher programs partnering with transportation providers to offer discounted or free transportation services for medical appointments;

c) Mobile service delivery extending medication distribution to remote communities on regular schedules, reducing required travel for stable participants;

d) Community-based distribution networks enabling medication access at local health outposts rather than centralized HIV clinics.

This hypothesis suggests that transportation support would be particularly effective for rural participants, those without stable employment, and individuals managing multiple health conditions requiring frequent healthcare visits. The substantial impact of employment status on adherence (β = 0.452, p < 0.001) documented in our empirical study and displayed in Table 4 suggests that addressing financial barriers to transportation would partially mitigate the adherence gap between employed and unemployed participants.

### 5.2. Decentralized medication distribution hypothesis

**Hypothesis 2:** Decentralized medication distribution networks that decouple routine medication refills from clinical monitoring appointments will significantly improve medication access while reducing healthcare system burden compared to conventional models requiring facility visits for all medication access.

This hypothesis addresses both the economic barriers (transportation costs, opportunity costs) and healthcare system factors (waiting times, privacy concerns) identified in our empirical investigation and presented in Tables 1 and 2. The current centralized distribution model requiring facility visits for all medication refills creates substantial barriers for participants managing competing responsibilities, particularly those in rural areas or with employment constraints. Recent implementation of decentralized drug distribution models in Eswatini and South Africa demonstrated that person-centered approaches adapting medication delivery to individual needs successfully decongest centralized facilities while maintaining treatment outcomes [35].

We propose that policy reforms enabling medication distribution through multiple channels would simultaneously enhance participant convenience while improving healthcare system efficiency. Specific policy mechanisms within this hypothesis include:

a)  Multi-month dispensing for stable participants, reducing required facility visits from monthly to quarterly or semi-annually;

b)  Community pharmacy partnerships enabling medication refills through local pharmacies rather than exclusively through HIV clinic pharmacies;

c)  Differentiated service delivery models tailoring appointment frequency and medication distribution channels to individual stability and needs;

d)  Mobile health technologies supporting remote medication management between less frequent clinical visits.

Large-scale randomized trials in Malawi and Zambia demonstrated that six-month multimonth dispensing is non-inferior to three-month dispensing for retention in care, with qualitative evidence showing high acceptability among both clients and providers [36,37]. The South African community pharmacy sector represents an untapped reservoir for HIV service delivery, with approximately 3,580 registered pharmacies that could expand access if appropriately integrated into treatment programs [38].

This hypothesis suggests that decentralized distribution would particularly benefit participants who are clinically stable, employed, and managing multiple responsibilities that compete with healthcare visits. By reducing facility congestion through fewer routine visits, this approach could simultaneously improve care quality for participants requiring more intensive clinical management.

### 5.3. Clinic efficiency and privacy hypothesis

**Hypothesis 3:** Redesigned clinic operations emphasizing reduced waiting times, enhanced privacy protections, and integrated (non-HIV-specific) service delivery will substantially improve treatment engagement compared to conventional HIV-specific service models.

This hypothesis addresses the counterintuitive findings regarding privacy and confidentiality presented in Table 2, where segregated clinical spaces demonstrated negative associations with adherence ($\beta = -.426$, $p < .001$) despite theoretical privacy benefits. These findings suggest that physically separating HIV services might inadvertently increase stigmatization through visible identification of participants accessing these specialized areas.

We propose that policy reforms focusing on operational efficiency and integrated service delivery would enhance both privacy and convenience, improving overall treatment engagement. Specific policy components include:

a)  Appointment systems with staggered scheduling to reduce waiting times and facility congestion;

b)  Integrated service models where HIV care occurs alongside general medical services without visible differentiation;

c)  Extended clinic hours accommodating employed participants' schedules and reducing peak-time congestion;

d)  Private consultation spaces within general healthcare settings rather than designated HIV-specific areas.

This hypothesis suggests that integrated service delivery would particularly benefit participants concerned about inadvertent status disclosure, younger participants managing identity concerns, and employed individuals with limited scheduling flexibility. The strong positive impact of comprehensive information provision on adherence ($\beta = .280$, $p < .001$) documented in Table 2 suggests that improved provider-participant engagement facilitated by more efficient systems could further enhance treatment understanding and management.

### 5.4. Economic support systems hypothesis

**Hypothesis 4:** Formalized income protection and nutritional support policies will significantly improve adherence outcomes compared to medical management alone, particularly for economically vulnerable participants experiencing employment instability, food insecurity, or competing financial priorities.

This hypothesis addresses the significant economic barriers documented in our empirical investigation and presented in Tables 1 and 4, including employment impacts ($\beta = 0.452$, $p < 0.001$) and broader economic constraints ($b = -2.0487$, $p < 0.001$). While clinical services have traditionally focused almost exclusively on medical management, the substantial economic determinants of adherence suggest that comprehensive HIV care must incorporate economic supports to achieve optimal outcomes. A systematic review examining poverty alleviation and HIV stigma concluded that livelihood interventions targeting poverty should be considered alongside ART to reduce stigma and improve treatment outcomes [28].

We propose that formalized policies addressing economic vulnerabilities would significantly enhance adherence capacity beyond what medical management alone could achieve. Specific policy components include:

a) Income protection programs for people living with HIV experiencing employment discrimination or health-related work limitations;

b) Nutritional support integrated with medication distribution, providing supplementary food packages during medication pickup;

c) Microfinance or vocational training opportunities specifically designed for HIV-affected households;

d) Health insurance reforms ensuring comprehensive coverage of HIV-related expenses beyond medication costs.

Evidence from ecological analyses across seven sub-Saharan African countries found that access to nutrition support services was the strongest predictor of non-attrition from HIV treatment among all facility-level factors examined [39]. Qualitative studies in Kenya demonstrated that people receiving ART enrolled in food support programs self-reported greater antiretroviral adherence, fewer side effects, greater ability to satisfy increased appetite, weight gain, recovery of physical strength, and resumption of labor activities compared to those without nutritional support [30]. The prospective longitudinal study in Senegal found that severe food insecurity predicted both loss to follow-up and poor adherence, with those experiencing persistent food insecurity showing eight-fold increased odds of poor adherence [40,41].

This hypothesis suggests that economic support would particularly benefit unemployed participants, those with limited social support networks, and individuals managing multiple household responsibilities with competing resource demands. The strong employment effect on adherence documented in our empirical study and presented in Table 4 suggests that interventions addressing economic stability could substantially improve treatment outcomes across multiple domains.

## 6. Testing the hypotheses

These policy hypotheses offer testable frameworks for evidence-based policy development. We propose specific approaches for evaluating each hypothesis through implementation science methods that balance experimental rigor with real-world applicability.

### 6.1. Phased implementation with evaluation

To test these hypotheses while simultaneously addressing critical adherence barriers, we recommend phased implementation incorporating rigorous evaluation components. This approach would enable evidence generation while providing immediate benefits to affected populations rather than delaying implementation pending complete evaluation.

For the Transportation Support Hypothesis, we recommend implementing different support mechanisms (subsidies, vouchers, mobile services) across comparable districts, measuring impacts on appointment attendance, medication possession ratios, and self-reported adherence. Cost-effectiveness analysis comparing different approaches would identify optimal strategies for scaling within resource constraints.

For the Decentralized Distribution Hypothesis, we recommend piloting different distribution models (multi-month dispensing, community pharmacy partnerships) with stable participants while maintaining conventional approaches for higher-need individuals. Comparative evaluation of medication possession ratios, viral suppression rates, and healthcare utilization patterns would identify most effective approaches for different participant subgroups. Implementation in Zimbabwe using community ART refill groups showed excellent retention rates at 12 months, with six-month dispensing proving non-inferior to three-month dispensing [37].

For the Clinic Efficiency Hypothesis, we recommend implementing staggered scheduling, extended hours, and integrated service delivery at selected facilities while maintaining conventional operations at comparable sites. Time-motion studies measuring waiting times, participant satisfaction assessments, and longitudinal adherence tracking would determine impacts on treatment engagement and clinical outcomes.

For the Economic Support Hypothesis, we recommend implementing different support packages (nutritional supplements, transportation vouchers, income protection) across comparable settings, measuring impacts on medication possession, appointment attendance, and clinical outcomes. Additional assessment of household economic status would determine whether interventions successfully address the underlying economic barriers to adherence.

### 6.2. Implementation considerations

Successful implementation requires attention to several critical factors beyond the specific interventions themselves:

**Stakeholder Engagement:** Policy development should incorporate perspectives from people living with HIV, healthcare providers, community organizations, and administrative officials to ensure interventions address actual rather than presumed barriers. The qualitative findings from our empirical study highlight the importance of incorporating lived experience in system design, as participants identified barriers that might not be immediately apparent to policymakers without direct ART experience.

**Context Adaptation:** While these hypotheses provide general frameworks, specific implementation should adapt to local contexts. Transportation challenges, for instance, differ substantially between urban and rural settings, requiring tailored approaches rather than one-size-fits-all solutions. The empirical evidence demonstrated significant variations in adherence barriers across different participant subgroups, suggesting the need for adaptable rather than rigid implementation models.

**Health System Capacity:** Implementation must consider existing health system capacity and potential unintended consequences. Decentralized distribution, for instance, requires robust inventory management systems to prevent stock-outs across multiple distribution points. Careful assessment of system readiness should precede implementation to prevent iatrogenic harm through well-intentioned but inadequately supported interventions.

**Sustainability Planning:** Implementation should include explicit sustainability mechanisms from inception rather than relying indefinitely on external funding sources. Cost-sharing approaches, integration with existing health system financing, and graduated transition plans should be incorporated into initial design to prevent program collapse when initial funding concludes.

## 7. Implications

### 7.1. Policy implications

These hypotheses have significant implications for HIV/AIDS policy development in Ghana and similar resource-limited settings. Most fundamentally, they suggest that effective adherence support requires multisectoral approaches extending beyond traditional healthcare boundaries to address the structural determinants of treatment engagement.

The Transportation Support Hypothesis implies that transportation infrastructure and healthcare accessibility should be explicitly linked in health policy development rather than treated as separate domains. Transportation ministries should participate in HIV program planning, incorporating healthcare accessibility into infrastructure development priorities. Similarly, HIV programs should incorporate transportation planning into service delivery models rather than assuming participant capacity to overcome mobility barriers independently.

The Decentralized Distribution Hypothesis challenges conventional facility-centered models that have dominated HIV care since ART introduction. This approach suggests that policy frameworks should distinguish between the clinical management and pharmaceutical distribution components of HIV care, potentially allowing separate optimization of each function rather than bundling them within a single service delivery approach.

The Clinic Efficiency Hypothesis implies that HIV-specific facilities, while initially necessary for program establishment, may create unintended stigmatization effects that undermine the very treatment engagement they aim to facilitate. This suggests that integration rather than exceptionalism should guide future HIV service development, particularly as treatment normalization progresses and clinical management increasingly resembles other chronic conditions.

The Economic Support Hypothesis most dramatically challenges conventional boundaries between health and social policy, suggesting that achieving optimal HIV treatment outcomes requires integration of income protection, nutritional support, and livelihood assistance within comprehensive care models. This implies need for formal coordination mechanisms between health ministries, social welfare agencies, employment offices, and agricultural programs rather than maintaining traditional sectoral silos.

### 7.2. Research implications

These hypotheses identify several priorities for future research. Comparative effectiveness studies examining different transportation support mechanisms would inform resource allocation decisions and identify optimal approaches for different contexts. Implementation science research investigating barriers and facilitators to decentralized distribution would support scale-up planning and identify necessary health system adaptations.

Research examining unintended consequences of integrated versus specialized HIV service delivery would clarify the privacy-stigma trade-offs suggested by our empirical findings. Economic evaluation comparing costs and benefits of comprehensive economic support programs versus conventional medical-only approaches would inform policy debates about resource allocation priorities.

Longitudinal studies tracking adherence trajectories among participants receiving different intervention combinations would illuminate potential synergies or antagonisms between different support mechanisms, informing development of optimized intervention packages rather than single-mechanism approaches.

### 7.3. Practice implications

For healthcare providers and facility managers, these hypotheses suggest several practice modifications even before formal policy changes. Clinic scheduling adjustments reducing waiting times and congestion require primarily operational changes rather than additional resources. Integration of HIV services within general medical services can begin through administrative reorganization of clinic spaces and scheduling.

Provider communication training emphasizing comprehensive information provision alongside respectful interpersonal engagement could improve adherence outcomes based on the strong associations documented in Table 2. Pharmacy services enhancement through additional pharmacist training or pharmacy technician deployment could strengthen medication support without requiring major infrastructure investments.

For community organizations and civil society groups, these hypotheses highlight advocacy priorities including transportation support formalization, economic protection policies, and nutritional assistance integration within HIV programs. Community-based distribution networks could be piloted through existing community health worker programs, potentially demonstrating feasibility and effectiveness before large-scale policy adoption.

## 8. Limitations

Several limitations warrant consideration when interpreting these policy hypotheses. First, the empirical evidence base derives from a single region within Ghana, potentially limiting generalizability to other contexts with different healthcare systems, cultural norms, or economic conditions. The cross-sectional design prevents causal inference despite statistical associations, requiring cautious interpretation of directional relationships between variables and outcomes.

The purposive sampling approach for qualitative interviews, while enabling in-depth exploration of adherence barriers among those experiencing treatment defaults, may not fully represent perspectives of consistently adherent individuals or those who have completely disengaged from care. Social desirability bias may have influenced both quantitative survey responses and qualitative interview accounts, particularly regarding sensitive topics like adherence behavior and stigma experiences.

Measurement limitations include reliance on self-reported adherence despite supplementation with pill counts and recall assessments, as objective measures like viral load testing were not uniformly available across all study sites. The quantitative survey instrument, while adapted from validated tools, may not have captured all relevant cultural or contextual factors influencing adherence in Ghana#39;s Ashanti Region.

The policy hypotheses themselves represent theoretical frameworks requiring empirical testing rather than evidence-based interventions, with actual effectiveness remaining uncertain until implementation and evaluation occur. Resource requirements and implementation feasibility vary substantially across different settings, potentially limiting applicability in severely resource-constrained contexts.

## 9. Conclusion

This article has presented four policy hypotheses for strengthening ART adherence in Ghana based on comprehensive empirical evidence from 2,000 people living with HIV and 20 qualitative interview participants. The evidence demonstrates that suboptimal adherence reflects complex interactions between economic constraints, healthcare system characteristics, and socio-cultural influences rather than simply individual motivation or knowledge deficits.

The Transportation Support Hypothesis addresses documented economic barriers preventing clinic attendance through formalized transportation subsidies, voucher programs, mobile service delivery, and community-based distribution networks. The Decentralized Distribution Hypothesis proposes medication access through multiple channels rather than exclusively facility-based distribution, simultaneously improving participant convenience and healthcare system efficiency.

The Clinic Efficiency and Privacy Hypothesis recommends integrated service delivery rather than segregated HIV-specific facilities, addressing counterintuitive findings that specialized HIV services may inadvertently increase stigmatization. The Economic Support Systems Hypothesis advocates for income protection and nutritional support integration within comprehensive HIV care, recognizing that medical management alone proves insufficient when economic barriers prevent consistent treatment engagement.

These policy hypotheses offer testable frameworks for evidence-based policy development that address structural determinants of adherence rather than focusing exclusively on individual behavior change. Implementation through

phased approaches with rigorous evaluation would enable evidence generation while providing immediate benefits to affected populations. Successful implementation requires multisectoral coordination extending beyond traditional health sector boundaries to incorporate transportation planning, social protection, employment policy, and nutritional assistance within comprehensive adherence support systems.

Moving from research evidence to policy action represents a critical step in translating scientific knowledge into population health improvements. These hypotheses provide specific, actionable frameworks for that translation, offering Ghanaian policymakers and their counterparts in similar contexts concrete starting points for developing evidence-based interventions addressing the multifaceted barriers that continue to undermine ART adherence and treatment effectiveness.

## Supporting information

**S1 Fig. Information Motivation Behaviour Model (Fisher & Fisher, 1992).**
(DOCX)

**S2 Fig. Health Belief Model (Vitalis, 2017).**
(DOCX)

**S1 File. De-identified quantitative survey data (N = 2,000, SPSS format).**
(ZIP)

**S2 File. Semi-structured interview guide for people living with HIV.**
(DOCX)

**S3 File. Semi-structured interview guide for healthcare providers.**
(DOCX)

**S4 File. Validated questionnaire for quantitative data collection.**
(DOCX)

**S5 File. De-identified qualitative interview excerpts by theme.**
(DOCX)

## Author contributions

**Conceptualization:** Emmanuel Oduro, Obed U. Lasim.

**Data curation:** Emmanuel Oduro, Victor Luckyboy Dzramado.

**Formal analysis:** Victor Luckyboy Dzramado.

**Methodology:** Emmanuel Oduro, Victor Luckyboy Dzramado.

**Project administration:** Emmanuel Oduro.

**Resources:** Emmanuel Oduro.

**Software:** Victor Luckyboy Dzramado.

**Supervision:** Emmanuel Oduro, Obed U. Lasim.

**Validation:** Emmanuel Oduro.

**Writing – original draft:** Victor Luckyboy Dzramado, Obed U. Lasim.

**Writing – review & editing:** Emmanuel Oduro, Victor Luckyboy Dzramado, Obed U. Lasim.

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
