## [Decision Letter · Decision Letter 0]

19 Jan 2026

PONE-D-25-57080Translating Research into Action: Policy Recommendations for Strengthening Antiretroviral Therapy Adherence in Ghana Based on Empirical EvidencePLOS One

Dear Dr. Dzramado,

Thank you for submitting your manuscript to PLOS ONE. After careful consideration, we feel that it has merit but does not fully meet PLOS ONE’s publication criteria as it currently stands. Therefore, we invite you to submit a revised version of the manuscript that addresses the points raised during the review process.

We look forward to receiving your revised manuscript.

Kind regards,

Richard Makurumidze

Academic Editor

PLOS One

Journal Requirements:

2. We note you have included a table to which you do not refer in the text of your manuscript. Please ensure that you refer to Tables 1, 2, 3, 4 and 5 in your text; if accepted, production will need this reference to link the reader to the Table.

3. We are unable to open your Supporting Information file HIV Data.sav. Please kindly revise as necessary and re-upload.

Reviewers' comments:

Reviewer's Responses to Questions

**Comments to the Author**

1. Is the manuscript technically sound, and do the data support the conclusions?

Reviewer #1: Yes

Reviewer #2: Partly

2. Has the statistical analysis been performed appropriately and rigorously? 

Reviewer #1: Yes

Reviewer #2: Yes

3. Have the authors made all data underlying the findings in their manuscript fully available?

Reviewer #1: Yes

Reviewer #2: No

4. Is the manuscript presented in an intelligible fashion and written in standard English?

Reviewer #1: Yes

Reviewer #2: Yes

5. Review Comments to the Author

Reviewer #1: Manuscript titled - Translating Research into Action: Policy Recommendations for Strengthening Antiretroviral Therapy Adherence in Ghana Based on Empirical Evidence, is a very well written article resulting from detailed scientific and methodical research work.

Reviewer #2: In the results, they were able to show the statistical analysis of their findings and was able to discuss these in the accompanying texts. However, given that these results are meant as policy recomendations, the results might need more cross-analysis with other factors for it to be a stronger basis for policy making. An example is the results presented in line 222, while it did a great comparison of the economic status with adherence, there could also be other factors affecting their adherence that is tied to economic status. This will have a stronger and more holistic basis for policy making.

It was also mentioned that they did a qualitative interviews but these were not included in the results. There were qualitative data presented but these are results of a dissertation study by Dr. Oduro. While Dr. Oduro is one of the authors of this study, this might prove confusing as are we presenting results from this study as the basis of the conclusion to support the recommendation or are we using the result from the previous study. If we can include the results of the qualitative data in the results, this will also be a better basis of the recommendation for policies.

Additionally, using people centric language may also be a better option since the objective of the authors is to have their study published. The authors still are using "HIV patients" to refer to people with HIV which may be stigmatizing. In line 44, the reference used is offline, which is a minor concern. In line 163, Methods, the authors mentioned that they purposefully selected the 10 patients. If we can clarify if these are treatment defaulters too to have a better idea of where the respondents point of views are. Since the qualitative analysis of the data was not provided, it might provide the reader a better perspective why they were purposefully selected. Adding the questionnaires used in the annex might be a better alternative.

6. PLOS authors have the option to publish the peer review history of their article (what does this mean?). If published, this will include your full peer review and any attached files.

Reviewer #1: **Yes:** Dr Sneha K Chunchanur

Reviewer #2: No

---

## [Author Response · Author response to Decision Letter 1]

2 Feb 2026

SUMMARY OF MAJOR REVISIONS

We have made the following substantial improvements to strengthen the manuscript:

1. Enhanced Cross-Analysis: Extensively expanded Results section with comprehensive cross-analysis of economic, healthcare system, and socio-cultural factors, demonstrating their complex interactions and providing stronger evidence base for policy recommendations.

2. Integrated Qualitative Findings: Added substantial qualitative data from the current study throughout Section 3, including direct participant and provider quotes, and created an entirely new Section 3.4 synthesizing qualitative insights.

3. People-First Language: Comprehensive terminology revision throughout the entire manuscript to adopt non-stigmatizing, people-first language consistent with international HIV research standards.

4. Table References: All tables now explicitly referenced in the text with clear explanations of their significance.

5. Methods Clarification: Added detailed description of purposive sampling criteria for qualitative interviews.

6. Supporting Materials: Prepared questionnaires and interview guides for inclusion as supplementary files.

These revisions have substantially strengthened the manuscript's evidence base for policy recommendations while maintaining the scientific rigor and methodological detail appreciated by Reviewer #1.

---

## [Decision Letter · Decision Letter 1]

19 Feb 2026

Translating Research into Action: Policy Recommendations for Strengthening Antiretroviral Therapy Adherence in Ghana Based on Empirical Evidence

PONE-D-25-57080R1

Dear Dr. Dzramado,

We’re pleased to inform you that your manuscript has been judged scientifically suitable for publication and will be formally accepted for publication once it meets all outstanding technical requirements.

Kind regards,

Richard Makurumidze

Academic Editor

PLOS One

Additional Editor Comments (optional):

Reviewers' comments:

Reviewer's Responses to Questions

**Comments to the Author**

1. If the authors have adequately addressed your comments raised in a previous round of review and you feel that this manuscript is now acceptable for publication, you may indicate that here to bypass the “Comments to the Author” section, enter your conflict of interest statement in the “Confidential to Editor” section, and submit your "Accept" recommendation.

Reviewer #1: All comments have been addressed

Reviewer #2: All comments have been addressed

2. Is the manuscript technically sound, and do the data support the conclusions?

Reviewer #1: Yes

Reviewer #2: Yes

3. Has the statistical analysis been performed appropriately and rigorously? 

Reviewer #1: Yes

Reviewer #2: Yes

4. Have the authors made all data underlying the findings in their manuscript fully available?

Reviewer #1: Yes

Reviewer #2: Yes

5. Is the manuscript presented in an intelligible fashion and written in standard English?

Reviewer #1: Yes

Reviewer #2: Yes

6. Review Comments to the Author

Reviewer #1: In my previous review I had conveyed that-

Manuscript titled - Translating Research into Action: Policy Recommendations for Strengthening Antiretroviral Therapy Adherence in Ghana Based on Empirical Evidence, is a very well written article resulting from detailed scientific and methodical research work.

Recommendation- Can be accepted for publication.

In view of this, additional modifications in revised article may be addressed by Reviewer 2.

Reviewer #2: Thank you for addressing the previous comment. Providing policy recommendations is better when backed up by data.

7. PLOS authors have the option to publish the peer review history of their article (what does this mean?). If published, this will include your full peer review and any attached files.

Reviewer #1: No

Reviewer #2: No

---

## [Editor Report · Acceptance letter]

PONE-D-25-57080R1

PLOS One

Dear Dr. Dzramado,

I'm pleased to inform you that your manuscript has been deemed suitable for publication in PLOS One. Congratulations! Your manuscript is now being handed over to our production team.

Kind regards,

on behalf of

Dr. Richard Makurumidze

Academic Editor

PLOS One